# Environmental Management System for the Analysis of Oil Spill Risk Using Probabilistic Simulations. Application at Tarragona Monobuoy

**Mariano Morell Villalonga ***, **Manuel Espino Infantes, Manel Grifoll Colls** and **Marc Mestres Ridge**

Laboratori d'Enginyeria Marítima (LIM), Universitat Politècnica de Catalunya (UPC-BarcelonaTech), C/Jordi Girona, 08034 Barcelona, Spain; manuel.espino@upc.edu (M.E.I.); manel.grifoll@upc.edu (M.G.C.); mmestresridge@gmail.com (M.M.R.)

* Correspondence: mariano.nicolas.morell@upc.edu

**Abstract:** Oil spill accidents during port operations are one of the main hydrocarbon pollution threats for coastal waters. Appropriate environmental risk assessment and pollution events management tools are needed to achieve sustainability and environmental protection in port activity. Recent developments in monitoring techniques and accurate meteo-oceanographic prediction systems have been implemented in many ports, providing tools for environmental management. A novel method based on meteo-oceanographic operational services, in conjunction with Monte Carlo experiments using an oil spill model, is implemented to perform probabilistic maps of potential pollution events. Tarragona port area was chosen as the study case for three reasons: it accommodates a hub of petrochemical industry, the availability of high-resolution wind and water current data, and previous studies at the area offer the possibility to check the results' accuracy. The interpretation of the impact probability maps reveals a specific pattern explained by the mean hydrodynamic conditions and the energetic north-westerly wind conditions. The impact probability maps may enhance efficiency in the environmental management of port waters and nearby coastal areas, reducing the negative impact of pollutant discharges.

**Keywords:** oil spill; environmental risk assessment; pollution events management; Tarragona port; SAMOA project; MEDSLIK model; Monte Carlo method

## 1. Introduction

The environmental pollution caused by port operational accidents has received increasing attention in the last decades, due to an increased environmental sensibility and shift towards blue growth economy concepts. In particular, pollution by hydrocarbons is relevant because of its frequency (they are present in approximately 57% of accidents involving chemical substances [1]) and their toxicity. The oil pollution of marine habitats is an issue, not only for researchers and environmentalists, but is also a main social and political concern, due to the serious impact of oil spills on marine life and on human activity, tourism, and the exploitation of the sea's resources.

The Marine Strategy Framework Directive, adopted at the European Union in 2008, requires member states to establish measures to achieve and maintain a good environmental status of marine waters. The Directive works on an ecosystem-based approach in the regulation and management of the marine environment, marine natural resources, and marine ecological services [2]. This approach requires the public administration and the private port operators to consider the potential effects of port activities on the marine environment in order to plan and manage port activity. This Directive

adds to the Water Framework Directive, which also takes into consideration coastal waters, setting a general scope on marine waters.

Port management policies need models in which the interactions of logistic and environmental factors can be considered, thus integrating the social, economic, legal, technical, and environmental demands together. In this context, environmental risk assessment instruments are meant to become the generalized tool for environmental management and decision-making for port authorities [3]. Several contributions faced risk port management using physical characterization of the oil spill and surrounding meteo-oceanographic conditions [4–8].

This management relies on a three-step process of: hazard identification, risk assessment, and risk management. In this sense, environmental risk assessment requires a description of hazards, the determination of the probability of impact, and the vulnerability of the environment, and thus derives the consequences from a hazard. This contribution focuses on the determination of the probability of impact using recent developments in monitoring techniques and accurate meteo-oceanographic information systems. A novel method, based on meteo-oceanographic operational services in conjunction with a Monte Carlo experiment of an oil spill model, is implemented to perform probabilistic studies of potential pollution events. The outcome focuses on the spatial distribution of impact probability of an oil spill in the dock or the monobuoy of the port of Tarragona, using a Monte Carlo method. We took advantage of the operational information available to use modelled wind and current conditions for the simulations. Additionally, the interpretation of the probability maps is carried out, linking with the meteo-oceanographic patterns of the region.

The paper is organized as follows. Section 2 introduces the study area, the risk management tool layout, the operational data source, and the oil spill model used. Section 3 shows the results of the simulations and a comparison with previous work on the same area. Section 4 presents a discussion on the design criteria for the risk management tool. Finally, in Section 5, the conclusions of the study are summarized.

## 2. Materials and Methods

### 2.1. Study Area

The port of Tarragona is located on the Mediterranean coast of Spain (Figure 1); approximate coordinates are: 1°14′ E, 41°05′ N. It is the main petrochemical port in the region, connected to one of the largest Spanish oil refineries, and also an important industrial and commercial port. Repsol Petróleo, SA, owner of the refinery, operates an oil terminal in the port, including a 1489 m long dock with mooring capacity for five vessels and a floating dock (monobuoy) for mooring and unloading larger vessels. This port is optimal for this study due to its activity, the availability of detailed meteo-oceanographic data from operational services, and the availability of previous oil spill environmental risk studies to compare against our results [4–6,9,10].

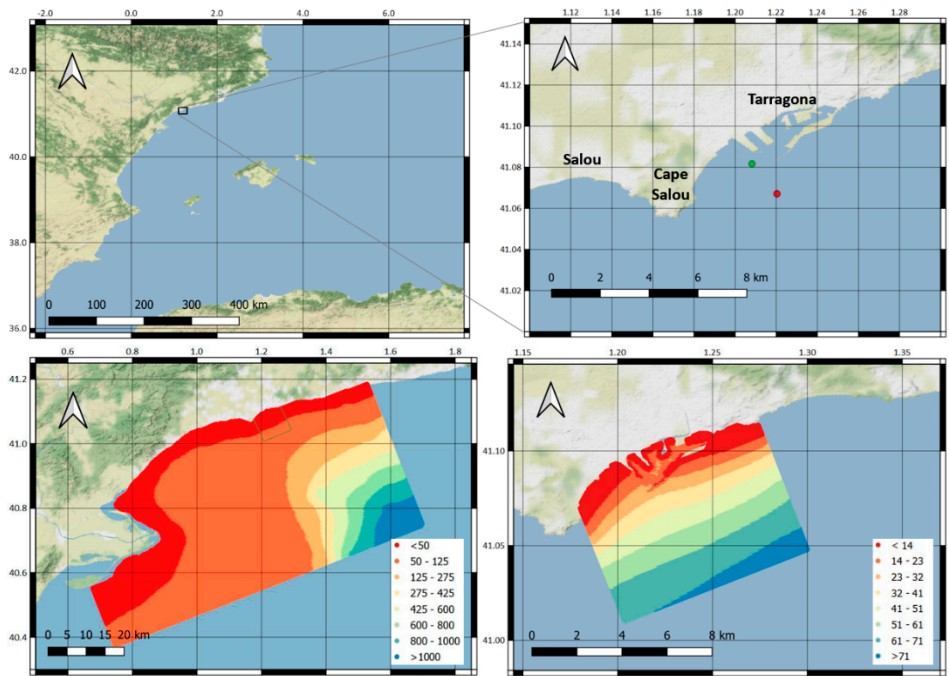

**Figure 1.** (**up-left**) Study location in the western Mediterranean; (**up-right**) situation of the spill points: dock (green dot) and monobuoy (red dot); (**low-left**) coastal and (**low-right**) port numerical domains of SAMOA data in Tarragona with bathymetry (in meters). Green box in the low-left figure represents the numerical boundary of the port domain nested in the coastal domain.

## 2.2. Meteo-Oceanographic Services

Wind and water current data for the model is obtained from the SAMOA system (in Spanish: Sistema de Apoyo Meteorológico y Oceanográfico de la Autoridad Portuaria). SAMOA is an initiative of the Spanish Public State Port Agency (Puertos del Estado) to provide port authorities with user-customized operational met-ocean information for harbor safety, environmental management, and operational decisions [11]. The SAMOA project provides hourly and daily values of meteo-hydrodynamic variables in the Tarragona Port area using two nested domains (see boundaries in Figure 1): Coastal domain (with a spatial resolution for currents of 350 m) and Port domain (70 m resolution). Wind data is derived from the Spanish Meteorological Agency (AEMET) forecast services, which use two operational applications of the high resolution limited area model (HIRLAM) model: one is the HNR, covering the Spanish territory, which has a 0.05° resolution and a forecast horizon of + 36 h, while the more extended regional euro-Atlantic ONR application has a 0.16° resolution and a forecast horizon of + 72 h [11].

The SAMOA project was inspired by the application of the regional ocean modelling system (ROMS) [12] over port and coastal domains in high-resolution meshes [13,14]. Water current data stored in the SAMOA is modelled using ROMS and initialized each day. The forecasts are systematically verified using three monitoring systems: (a) the buoy of Tarragona (REDEXT code 2720, location 1.47 E-40.68 N); (b) the mareograph of Tarragona (REDMAR code 3756, location. 1.21 E-41.08 N); and (c) the high frequency radar system of Delta de l'Ebre (with three stations at Vinaroz, Alfacada, and Salou) [15]. SAMOA provides hourly averaged results, so this frequency was high enough for our probabilistic method when spatial scope was hundreds of meters (or higher resolution) and dispersion effect was considered.

## 2.3. Probabilistic Risk Management

Oil spill hazard can be described under source-pathway-receptor-consequence (S-P-R-C) methodology [8], in which the analysis of the potential pathway between source and receptor is

a critical point. In this sense, environmental risk management tools require the hydrodynamic information of the receptor domain [16]. Several types of environmental risk management instruments have been postulated in recent years in order to mitigate the environmental impact of port activities. These instruments can be classified into nine groups, according to Reference [17], based on their analytical approach: even tree analysis, failure mode and effects analysis, fault tree analysis, risk maps, scenario analysis, Bayesian belief networks, decision tree, bow-tie analysis, and cause-consequence analysis. The support method used may be classified into four groups: analytical hierarchy process, fuzzy theory, evidential reasoning, and simulation methods. This work was directed by the risk map approach, supported by simulation methods. Several examples of such instruments can be found in References [4–6,12,14,16,18]. They vary depending mainly on the perspective adopted, the information available, and the purpose pursued. In any case, all these tools will be articulated by combining a set of constituent elements within an operational layout and the corresponding decision-making criteria.

In general, the common layout of any of the mentioned tools have the same flow chart: information or input variables, one or several numerical simulation models, and one or several outputs that can be used to support the decision making. This modular structure allows improvements in any of its integrating elements to be incorporated into the system. Our contribution focuses on a tool schematized in Scheme 1, whose purpose is the elaboration of probability maps associated with oil spills in the oil transfer facilities of the port of Tarragona. The investigation focuses on the application of an oil spill modelling, the statistical application of meteo-oceanographic operational products, and the physical interpretation of the model output.

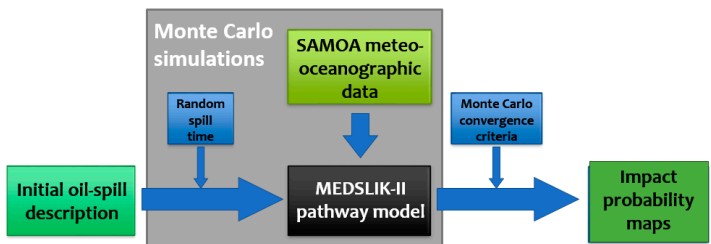

**Scheme 1.** Layout of the environmental management tool for accidental spills in the oil transfer facilities of the port of Tarragona.

The environmental management tool is based on a set of Monte Carlo iterations using oil spill simulations obtained from an upgraded version of the MEDSLIK-II model. MEDSLIK-II is an open source Lagrangian model, developed in 2013, to simulate the transport and aging of the slick produced by a spill of oil or a derivate in the sea [19,20]. Oil transport is governed by the water currents and the wind and dispersed by turbulent fluctuation components that are parameterized with a random walk scheme. In addition, the model takes in consideration the oil spill evolution due to various physical and chemical processes that transform the oil (i.e., evaporation, emulsification, dispersion in water column, adhesion to coast). MEDSLIK-II is the pathway model chosen for the oil spill module in the SAMOA II project (currently in development). It is also the reference oil spill model adopted by the MONGOOS network, the EMODNET Mediterranean Sea Check Point, and MEDESS-4MS European projects. MEDSLIK-II has also been used in several scientific contributions, e.g., in References [21–24].

The upgraded version used in this article was elaborated in the framework of the CEASELESS H2020 EU research project [25]. The modifications were focused in adapting both forward simulation, to determine the evolution of the spot from a given spill point, and the backtracking procedure, to determine its origin from the point where the spot had been detected.

For the initial oil spill modeling, the premises postulated in the Maritime Interior Plan of the Repsol Terminal of the port of Tarragona (written in 2009) were used. According to this document, a characteristic spill of 5.4 Tn of crude oil (the amount of crude spilled had no incidence on the maps obtained as the impact of pollution was considered without any concentration threshold) during a 5 min period was considered. In this sense, initial conditions were implemented considering the 5 min

after the accident, simplifying the initial speed of the spill and the possible movement of the discharge point, etc. In consequence, an initial 10 000 $m^2$ square spilled area was considered at the beginning of the simulation.

The model was forced by the wind and water current fields that were introduced in the upgraded model in either 2 or 3 dimensions; in this case 2-dimensional water currents were used. The wind and water current data used were the historical numerical results obtained from the SAMOA Project [11]. The output of the oil spill model was the position of the tracer particles used to simulate the oil-spill at different time steps. Thus, the results of different simulations were added in order to obtain the probability maps on the superposition of tracer particles of all the simulations considered at the chosen time steps.

The statistical method adopted to determine the spatial distribution of the probability of impact was the Monte Carlo algorithm. The Monte Carlo algorithm was carried out by simulating a set of spills characterized by a random initial spill time within the simulation period that spanned between October 2017 and September 2018. That is, multiple random combinations of days and hours, representing initial spills that were generated within this period. Then, the evolution of hypothetic spills of the exposed characteristics occurring at these random times were simulated in MEDSLIK-II. To establish this simulation period, the temporal continuity of meteo-oceanographic operational data was analyzed to provide one continuous year of data that allowed us to have an even distribution of the simulations along all seasons. Oil spill simulations were carried out in the two available domains, considering alternatively the dispersion effect. Thus, four types of numerical experiments were designed, as shown in Table 1.

**Table 1.** Numerical experiment classes.

| Domain | With Dispersion (D) | Without Dispersion (N) |
|:---:|:---:|:---:|
| Port (P) | PD | PN |
| Coast (C) | CD | CN |

Numerical experiments for the four experiment classes were carried out considering two spill point options: the monobuoy and the dock (see locations at Figure 1). The dock oil-spill location corresponded to its final section of the dock. The model parameters used for each of these four types of experiments and for both spill points are summarized in the Table 2.

**Table 2.** Model parameters for numerical experiments.

| Parameter | PD | CD | PN | CN |
|:---:|:---:|:---:|:---:|:---:|
| Steps/hour [1] | 82 | 10 | 82 | 10 |
| Interval [2] | 0.05 | 0.1 | 0.05 | 0.1 |
| Parcels [3] | 10 | 10 | 1 | 1 |
| Hz diffusivity [4] | 10 | 10 | 0 | 0 |
| Duration [5] | 4 | 8 | 4 | 8 |

[1] Number of time steps per hour used for slick computation. [2] Interval for output (h). [3] Number of parcels used to model diffusion and dispersion. [4] Horizontal diffusivity ($m^2$/s). [5] Duration of computation from spill start (h).

The number of time steps per hour was determined by computation requirements. The output interval was established according to the scale of the probability map grid. The number of parcels was 1 when dispersion was neglected and 10 when the dispersion was considered. The value adopted for horizontal diffusivity was obtained from the literature [26,27] and a sensitivity analysis was carried out. Finally, the oil spill duration was established according to the size of each domain.

Added to Monte Carlo simulations, numerical experiments were carried out for specific hours and months within the simulation period in order to analyze variations in the distribution of the probability

of impact of specific temporal scales (e.g., hourly or seasonal). The results from these non-Monte Carlo numerical experiments were not used for the impact probability maps, but as an interpretation tool.

The results obtained in the port with dispersion (PD) and coast with dispersion (CD) experiments have been used for the generation of impact probability maps using a two-step process. In the first step, particle-count maps were generated by defining a mesh on the domain and obtaining for each cell in the mesh the total count of tracer particles that were in that cell at any step of any simulation. In the second step, impact probability maps (IPMs) were obtained by normalizing the corresponding particle count map, that is, by dividing the value in each cell by the maximum value that corresponds to the cell that contains the initial spill zone. This way, IPMs showed the probability of presence of tracer particles in each cell at any time for simulation lasting 8 h. The cell size used was $100 \times 100$ m. Probability was defined only in the area where numerical convergence of the Monte Carlo simulations was achieved. For visualization purposes, a logarithmic probability scale was chosen.

## 3. Results

Figures 2 and 3 show the IPMs for spills in the dock and monobuoy, respectively, evaluated on the coastal numerical domain (see Figure 1). Comparison of these maps shows that potential spills occurred in the monobuoy can affect significantly larger areas than spills occurred in the dock. This difference is particularly relevant in the east direction, in which the port constitutes a significant barrier for spills released from the dock. In the south direction, the spill can reach approximately 20% further from the monobuoy than from the dock, apart from the fact that the monobuoy is about 1.5 km south from the dock. In the southwest direction, a spill from the monobuoy can reach approximately 50% further than the spill from the dock.

Figures 4 and 5 show the IPMs for spills in the dock and monobuoy, respectively, computed on the port numerical domain. In order to avoid the effect of the numerical domain boundary, these maps were defined for a probability of impact higher or equal than 1.5625%, although convergence has already been achieved at lower probabilities (see considerations about convergence in the Discussion section).

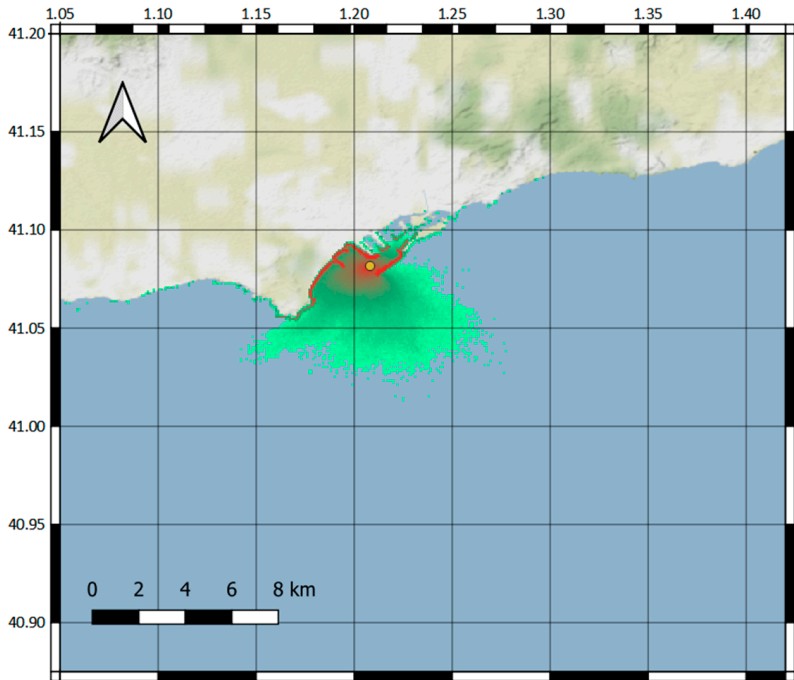

**Figure 2.** Impact probability map (IPM) for spills in the dock (green dot), evaluated on the coastal domain.

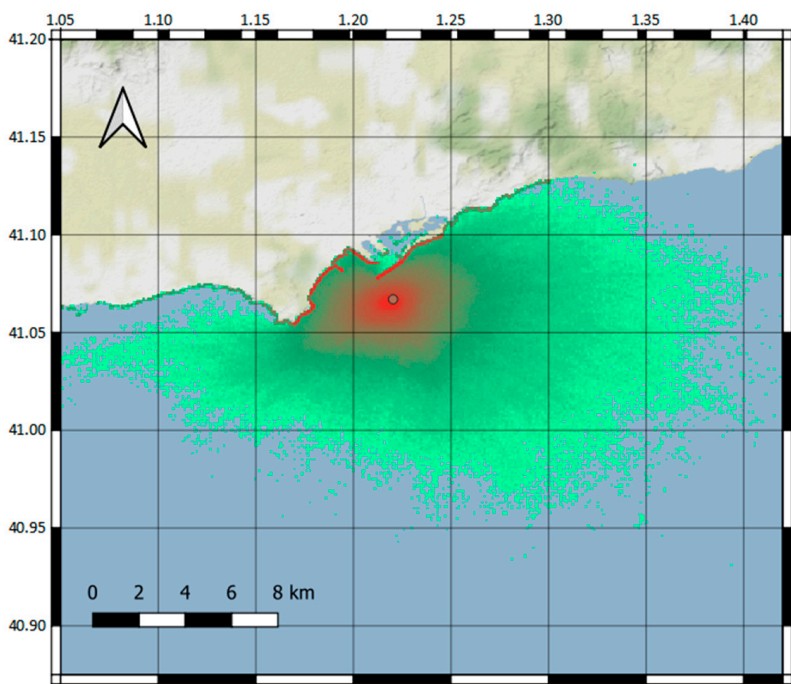

**Figure 3.** IPM for spills in the monobuoy (brown dot), evaluated on the coastal domain.

Comparison of these maps show again that spills occurred in the monobuoy can potentially affect a significantly larger area than spills occurred in the dock. The relative difference is higher in this case: 55% in the south direction and 85% in the southwest direction, and, again, in the east direction, the port constitutes a significant barrier for spills in the dock.

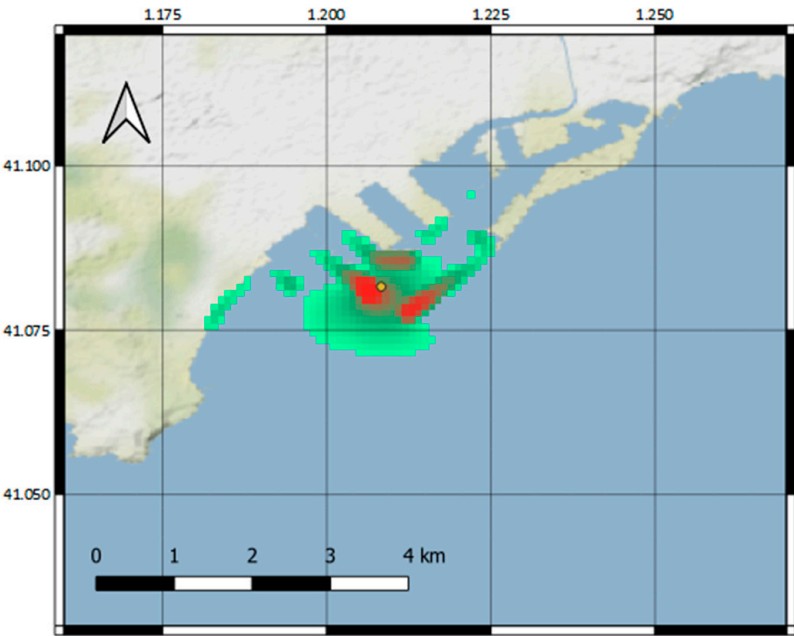

**Figure 4.** IPM for spills in the dock (yellow dot), evaluated on the port domain.

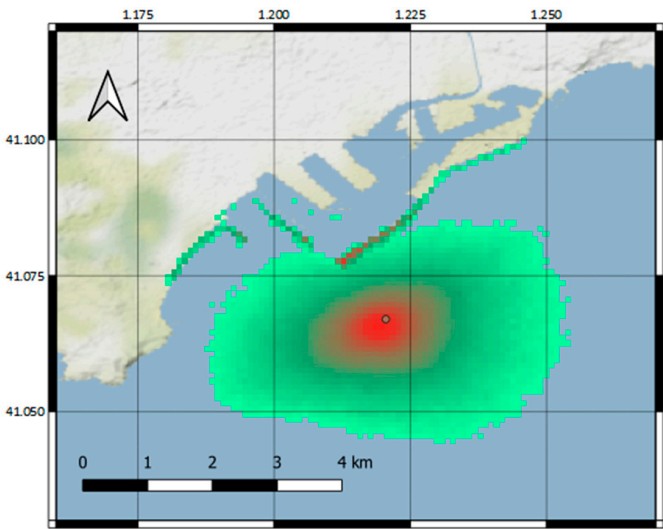

**Figure 5.** IPM for spills in the monobuoy (brown dot), evaluated on the port domain.

## 4. Discussion

### *4.1. Numerical Resolution Comparison*

Previous IPM results have allowed us to evaluate the impact of the numerical resolution of the oil-spill in coastal areas. In these computational experiments, the Port model used a larger resolution to describe the water current in comparison to the Coastal model and expected a more accurate solution in the first case. However, Port domain boundary is a significant limitation to be taken into account. IPM comparison suggests similar maps in both domains for the case of the monobuoy, although this may not be obvious when comparing Figures 3 and 5 because of the difference in ambit extension and probability representation scale. On the other hand, divergences between IPMs computed using Port and Coastal domains suggest differences of oil spill numerical solutions in the function of the hydrodynamic numerical resolution. The reason is that it seems associated with the hydrodynamics described in the port entrance (i.e., near the oil handling dock), which is quite complex, and there is a significant loss of information in its representation on the coarser mesh (see Figure 6).

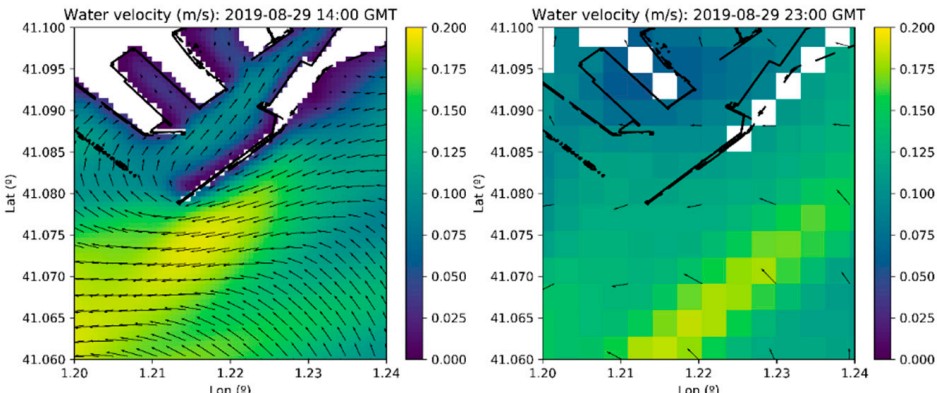

**Figure 6.** Surface water velocity obtained from SAMOA products (**left**: Port model, **right**: Coastal model).

### *4.2. Temporal Variability and Hydrodynamic Considerations*

The temporal variability of the IPM was investigated in order to determine the prevalent hydrodynamic pattern. In this sense, monthly IPMs were analyzed and compared with the mean water circulation. Variation on IPMs for different seasons could not be clearly established, as the difference

between different seasons was apparently no greater than differences observed between consecutive months. The differences detected were not considered significant given that only 1 year of data was analyzed, although, for the same reason, the existence of seasonal variations cannot be ruled out.

The daily variability was also analyzed, obtaining IPMs for oil spills released at different hours. The oil-spill hours considered were 04:00, 10:00, 16:00, and 22:00 GMT. In this analysis, a significant variation was found.

- For spills released at 04:00, the high probability area was slightly displaced away from the coast and the average distance reached was slightly lower.
- For spills released at 10:00, the high probability area was near the coast with a high proportion of particles trapped on the shoreline and the average distance reached was quite a lot lower.
- For spills released at 16:00, there was a wider and more uniform distribution with an average proportion of particles trapped on the coast and the average distance reached was higher.
- For spills released at 22:00, a wider and more uniform distribution (though not as much as at 16:00 spills) was observed and the average proportion of particles trapped on the shore and the average distance reached was higher.

The analysis of water current data from the SAMOA project shows an averaged water circulation southwestward, consistent with the shape of the IPM. Figure 7 shows the surface water current averaged for the year 2019, for which the maximum velocities were obtained in the vicinity of the monobuoy. This hydrodynamic pattern is common in the regional water circulation in the inner shelf, where hydrodynamics is modulated by remote sea level gradients and regional winds [28,29]. Overlapped to mean water circulation, measured wind data from a meteorological station at Tarragona Port shows how the most energetic wind conditions correspond to the NW component (Figure 8). In this sense, a dominant northwesterly component during winter and fall occurs, according to previous studies based on long-term measurements [30–32]. This would explain the offshore principal direction of the IPM that was consistent with the offshore winds. The additional sea-breeze cycle during summer may provide offshore water flow. However, opposite to onshore flow, offshore flow was neglectable in comparison to long-shelf circulation (see Figure 7). Therefore, the spatial variability to IPMs shown previously presumably corresponds to the NW wind component and southwestward averaged water circulation in the surface.

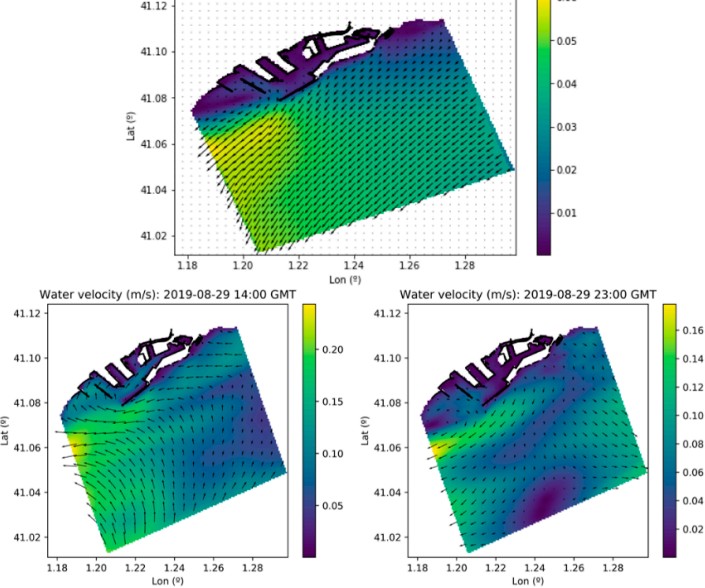

**Figure 7.** (**upper**) Averaged water current from the Port model during 2019. Water current fields during sea-breeze event: 2019-08-29 14:00 GMT (**lower left**) and 2019-08-29 23:00 (**lower right**). Water current fields were obtained from the SAMOA project.

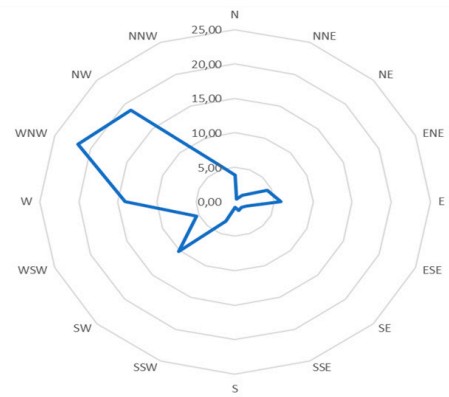

**Figure 8.** Wind direction probability distribution for strong winds (≥ 7.5 m/s). Data obtained from the Spanish Port Agency (*Puertos del Estado*) during the February 2002–January 2007 period.

*4.3. Convergence Considerations*

A key topic when using the Monte Carlo method is the convergence assessment criteria. As any inference based on the Monte Carlo method, the output relies critically upon the assumption that the Markov chain has achieved a steady state (i.e., converged) [33].

The first step to consider is whether convergence can be achieved at the same time for all the cells in the IPM. With this method, each cell after a given number of simulations will have a different degree of steadiness. The map does not converge as a whole, but each cell does converge after a certain number of simulations. For a given number of simulations, the map will show an irregular area of converged cells and non-converged cells. Therefore, we established a cell convergence criterium to fulfill the following conditions:

- The criteria should be based on relative error in probability, instead of absolute, as it will have to work consistently for different probability values.
- If the criterium takes into consideration the probability value, it must be evaluated on the estimate probability obtained at any given number of simulations.
- A criterium evaluation with low evaluation cost will be preferred so it can be evaluated after each simulation without a high increase in the time needed for calculation.

To propose a criterium, we compared the number of simulations with the probability obtained for each cell: the number of spill hits in the cell divided by the total number of simulations. This comparison showed a spiked profile (see Figure 9) with a spike at each simulation, in which the spill hit the cell. Spike heights decreased as the number of simulations increased.

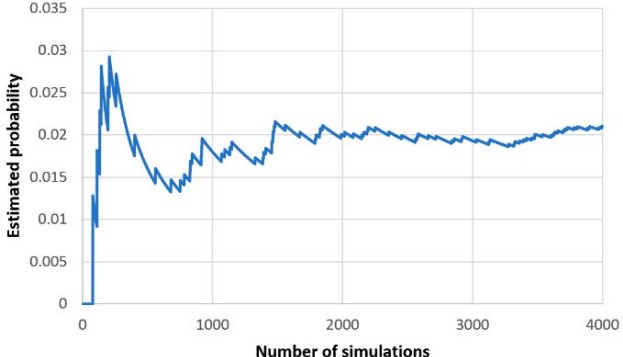

**Figure 9.** Comparison example between number of simulations and the estimated impact probability for a cell.

One reasonable cell convergence criterium that fulfilled the previous conditions was as follows: Convergence was achieved when the expected spike height for a hit was less than a chosen fraction of

the probability obtained. This criterium could be considered in terms of the number of hits needed for convergence. This number depended on hit probability as the quotient between expected spike height, and probability obtained increased as probability decreased (see Table 3). This value decreased to a threshold of the inverse of the number of hits. These values were obtained by taking into account the properties of the binomial probability distribution. The binomial is a discrete distribution that applies to the number of successes in a sequence of independent experiments. That is the case of Monte Carlo experiments when data from one experiment is not taken into account for other experiments.

**Table 3.** Quotient between expected spike height and probability obtained for different combinations of probability and number of hits.

| Probability | 2 Hits | 5 Hits | 8 Hits | 9 Hits | 10 Hits | 11 Hits |
|:---:|:---:|:---:|:---:|:---:|:---:|:---:|
| 50% | 0.333 | 0.111 | 0.067 | 0.059 | 0.053 | 0.048 |
| 25% | 0.429 | 0.158 | 0.097 | 0.086 | 0.077 | 0.070 |
| 10% | 0.474 | 0.184 | 0.114 | 0.101 | 0.091 | 0.083 |
| 1.0% | 0.498 | 0.198 | 0.124 | 0.110 | 0.099 | 0.090 |
| ≤ 0.10% | 0.500 | 0.200 | 0.125 | 0.111 | 0.010 | 0.091 |

In this work, a criterium based on an absolute number of hits, equal to 10, was chosen. This criterium set the quotient between expected spike height and probability obtained to the ratios shown in Table 3. This criterium is as demanding as considering one tenth of the probability obtained when the probability was 11.11% or lower, and more demanding with higher probabilities (e.g., with 50% probability, 6 hits would be enough to reach one tenth of the ratio). The actual computational cost of this simplification is quite low as the expected number of simulations to get a hit is the inverse of the probability.

### 4.4. Comparison with Previous Works

Potential oil spills from the Tarragona monobuoy have been analyzed previously in several contributions [4–6]. These works also show an aggregate pattern, and the main directions were E, ESE, and WSW. The IPMs for the monobuoy spill point on both domains (Figures 3 and 5) were consistent with these contributions showing the probability shape that was elongated on these three directions (Figure 10). Reference [4] uses input meteo-oceanographic conditions based on numerical modelling of characteristic scenarios. The TESEO oil spill model [18] shows these prevalent directions in potential oil spills from the modelling of characteristic scenarios. The IPMs for both spill points on the coastal domain (Figures 2 and 3) were also consistent with the mentioned contributions [9,10], which point out the protection provided by cape Salou to the city of Salou and the nearby beaches.

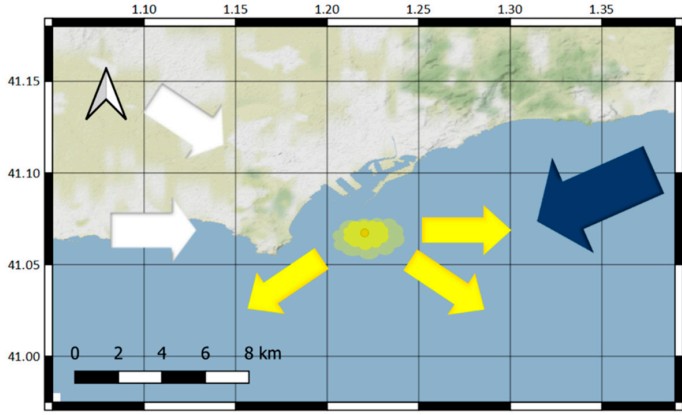

**Figure 10.** Oil transport pattern for spills at the monobuoy. Main directions are represented with yellow arrows. The blue arrow represents the main water current. White arrows represent the main wind directions.

*4.5. Port Management Applications*

The statistical methodology based on the Monte Carlo method offers port managers a powerful tool for oil spill risk management, which will result in better compliance with the objectives set by the Marine Strategy Framework Directive. The implementation of this system is facilitated by the implementation of meteo-oceanographic operational models in port areas (e.g., SAMOA). Therefore, the possibility of implementing tools to define the IPMs on the operating models is an added value with low computational effort in comparison to the operational meteo-oceanographic model itself.

Additionally, this methodology has the advantage of automatically improving the output when more historical wind and current data are stored. In consequence, its reliability will grow dynamically without the need to periodically redesign. This advantage also makes it suitable for port environments, where the knowledge of local meteo-oceanographic conditions is limited but operational models are being implemented.

## 5. Conclusions

In this paper, a probabilistic method to obtain IPMs using Monte Carlo simulations is presented. The implementation of the method at oil facilities in Tarragona Port suggests that the IPM is a valid tool for the environmental management in ports. In this case, the IPMs are consistent with the meteo-oceanographic characteristics of the region: south-westward averaged water circulation and NW energetic wind events. The potential of this method will grow in concordance with the development of meteo-oceanographic operational systems models in ports and coastal areas. During the tool design, a compromise has to be reached for the scope and scale of the study, taking into account the available meteo-oceanographic information and the model requirements. Expert judgment will be necessary for analysis of low probability levels in areas with limited data. The analysis of these situations will determine adequate strategies to overcome the limitations being an interesting line for future research.

**Author Contributions:** Conceptualization, M.M.V., M.E.I. and M.G.C.; methodology, M.M.V., M.E.I. and M.G.C.; software, M.M.V., M.G.C. and M.M.R.; validation, M.E.I. and M.G.C.; formal analysis, M.M.V., M.G.C. and M.M.R.; investigation, M.M.V. and M.M.R.; resources, M.E.I. and M.G.C.; data curation, M.M.V. and M.M.R.; writing—original draft preparation, M.M.V., M.E.I. and M.G.C.; writing—review and editing, M.M.V., M.E.I. and M.G.C.; visualization, M.M.V. and M.G.C.; supervision, M.E.I. and M.G.C.; project administration, M.M.V., M.E.I. and M.G.C.; funding acquisition, M.E.I. and M.G.C. All authors have read and agreed to the published version of the manuscript.

**Funding:** This research received funding from SAMOA2 project under agreement with Puertos del Estado.

**Acknowledgments:** The authors are grateful for the data provided by Puertos del Estado, Repsol Petróleo and Port de Tarragona. This work received funding from SAMOA 2 project under agreement with Puertos del Estado. The authors also want to thank the Secretaria d'Universitats i Recerca del Dpt. d'Economia i Coneixement de la Generalitat de Catalunya (ref. 2014SGR1253), who supported our research group.

**Conflicts of Interest:** The authors declare no conflict of interest. The funders had no role in the design of the study; in the collection, analyses, or interpretation of data; in the writing of the manuscript, or in the decision to publish the results.

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
