# Peer review of "Environmental Management System for the Analysis of Oil Spill Risk Using Probabilistic Simulations. Application at Tarragona Monobuoy"

_jmse, doi:10.3390/jmse8040277_

Round 1

Reviewer 1 Report

1. In line 86, the cars write: "Several types of environmental risk management instruments can be considered." What kind of instruments?
2. On what basis was the size of the simulated leakage determined? Have other spillage sizes been considered?
3. The lierature review lacks an overview of existing models/methods used to calculate the probability of pollution. I think this should be included in the literature review.

Author Response

Dear Reviewer,

We sincerely thank you for your valuable and constructive comments on our manuscript entitled “Environmental management system for the analysis of oil spill risk using probabilistic simulations. Application at Tarragona monobuoy” (ID: jmse-760245). Those comments and suggestions have improved the quality of the manuscript immensely, as well as providing important guiding significance to further research. We have analysed your comments carefully and several corrections have been included which we hope meet with your approval. The corrections have been introduced in the new version of the manuscript.

Below are the answers or explanations, written in blue, after each comment.

[Comment 1] In line 86, the cars write: "Several types of environmental risk management instruments can be considered." What kind of instruments?

[Response] We have rewritten the introduction and in section 2.3 and additional information has been added about environmental risk management methodologies as well as several references on such instruments.

[Comment 2] On what basis was the size of the simulated leakage determined? Have other spillage sizes been considered?

[Response] Section 2.3 has been rewritten and expanded to explain more clearly the process followed and the leakage simulated and how this assumption has no significative incidence on the maps obtained. Corresponding paragraph is in lines 145-152.

[Comment 3] The lierature review lacks an overview of existing models/methods used to calculate the probability of pollution. I think this should be included in the literature review.

[Response] Introduction and section 2.3 have been expanded with additional information on models andmethods used to calculate the probability distribution of pollution impact.

We appreciate for your warm work earnestly, and hope that the corrections will meet with approval.

Once again, thank you so much for your comments and suggestions!

Reviewer 2 Report

General remark:

The introduction is very general, and the literature overview is limited, previously developed methodologies are not adequately treated; and please briefly explain how your approach is overcoming existing limitations…

Please clarify:

Which upgraded functionality of MEDSLICK-II software was beneficial for your needs; i.e., achieving enhanced modelling of released spill stain in the “vicinity” of the port area?

The sentence starting with the row 116 is not very clear, can you please explain?

Author Response

Dear Reviewer,

We sincerely thank you for your valuable and constructive comments on our manuscript entitled “Environmental management system for the analysis of oil spill risk using probabilistic simulations. Application at Tarragona monobuoy” (ID: jmse-760245). Your remark and clarifications have improved the quality of the manuscript immensely, as well as providing important guiding significance to further research. We have analysed your comments carefully and several corrections have been included which we hope meet with your approval. The corrections have been introduced in the new version of the manuscript.

Below are the answers or explanations, written in blue, after each comment.

[General remark] The introduction is very general, and the literature overview is limited, previously developed methodologies are not adequately treated; and please briefly explain how your approach is overcoming existing limitations…

[Response] We have rewritten the introduction, addressing it more precisely to the subject of our paper. Additional information has been added in the introduction and in section 2.3 about risk management methodologies. An explanation on how our approach is overcoming existing limitations is also included in the introduction and referred in the conclusions.

[Clarification 1] Which upgraded functionality of MEDSLICK-II software was beneficial for your needs; i.e., achieving enhanced modelling of released spill stain in the “vicinity” of the port area?

[Response] An explication on functionality of MEDSLIK-II model and the reasons why it suits our needs have been included in Section 2.3.

[Clarification 2] The sentence starting with the row 116 is not very clear, can you please explain? [An instant spill has been considered since the duration considered in the plan, 5 minutes, is negligible compared to the simulation interval; having verified that this hypothesis does not significantly alter the probability maps obtained.]

[Response] This sentence and the whole paragraph have been modified as to make them clearer. Corresponding paragraph is in lines 145-152.

We appreciate for your warm work earnestly, and hope that the corrections will meet with approval.

Once again, thank you so much for your comments and suggestions!

Reviewer 3 Report

In the paper used MEDSLIK-II model. There are many similar models in the World that allow calculating the transformation and dynamics of oil in the sea and based on the same physical and chemical mechanisms. The principles of the model’s operation are not disclosed in the text of the paper, but only general words are given (copied from http://www.medslik-ii.org/model.html). In scientific work, one should be critical in choosing a model. You should give an overview of several of the most famous models and a comparison of their capabilities, as well as justify the choice of your model and show what its advantage over many others. Now the choice of MEDSLIK-II does not look convincing.

It is advisable to describe in more detail the wind and current data [9] in Section 2.2. Meteo-oceanographic services. What is the data? Are direct measurements used (ADCP, Anemometers, etc.)? what is the accuracy of measurements and where are the sensors installed in the area under consideration? Is satellite information used and how is it validated?

How is information processed and prepared for using in modeling? It is necessary to evaluate how the non stationarity of the wind field and current will affect the probability distribution (for example, vortex structures in the water area, atmospheric fronts, etc.).

If possible, bring a modeling scheme indicating "external" data sources. For two-dimensional probability distribution maps, it is also recommended to bring their sectional drawing in characteristic directions. Figures do not have a grid and north arrow.

The system provides data on wind and flow with an averaging of 1 hour. Wind variability is usually much stronger than current variability, what justifies the use of such low frequency wind data as initial conditions? It is necessary to give a detailed explanation that in this case this is permissible and does not give big errors.

Author Response

Dear Reviewer,

We sincerely thank you for your valuable and constructive comments on our manuscript entitled “Environmental management system for the analysis of oil spill risk using probabilistic simulations. Application at Tarragona monobuoy” (ID: jmse-760245). Those comments and suggestions have improved the quality of the manuscript immensely, as well as providing important guiding significance to further research. We have analysed your comments carefully and several corrections have been included which we hope meet with your approval. The corrections have been introduced in the new version of the manuscript.

Below are the answers or explanations, written in blue, after each comment.

[Comment 1] In the paper used MEDSLIK-II model. There are many similar models in the World that allow calculating the transformation and dynamics of oil in the sea and based on the same physical and chemical mechanisms. The principles of the model’s operation are not disclosed in the text of the paper, but only general words are given (copied from http://www.medslik-ii.org/model.html). In scientific work, one should be critical in choosing a model. You should give an overview of several of the most famous models and a comparison of their capabilities, as well as justify the choice of your model and show what its advantage over many others. Now the choice of MEDSLIK-II does not look convincing.

[Response] An explication on functionality of MEDSLIK-II model and the reasons why it suits our needs have been included in Section 2.3.

[Comment 2] It is advisable to describe in more detail the wind and current data [9] in Section 2.2. Meteo-oceanographic services. What is the data? Are direct measurements used (ADCP, Anemometers, etc.)? what is the accuracy of measurements and where are the sensors installed in the area under consideration? Is satellite information used and how is it validated?

[Response] An explication on functionality of MEDSLIK-II model and the reasons why it suits our needs have been included in Section 2.3.

[Comment 3] How is information processed and prepared for using in modeling? It is necessary to evaluate how the non stationarity of the wind field and current will affect the probability distribution (for example, vortex structures in the water area, atmospheric fronts, etc.).

[Response] Section 2.3 has been rewritten and expanded to explain more clearly how wind and current data have been used. Section 2.2, on the SAMOA project and how wind and water current fields are obtained, has also been expanded and clarified.

[Comment 4] If possible, bring a modeling scheme indicating "external" data sources. For two-dimensional probability distribution maps, it is also recommended to bring their sectional drawing in characteristic directions. Figures do not have a grid and north arrow.

[Response] Scheme 1 has been modified as well as figures 1 to 6.

[Comment 5] The system provides data on wind and flow with an averaging of 1 hour. Wind variability is usually much stronger than current variability, what justifies the use of such low frequency wind data as initial conditions? It is necessary to give a detailed explanation that in this case this is permissible and does not give big errors.

[Response] Section 2.2 has been rewritten and this question has been included in it.

We appreciate for your warm work earnestly, and hope that the corrections will meet with approval.

Once again, thank you so much for your comments and suggestions!

Round 2

Reviewer 3 Report

I am sincerely grateful to the authors for the work done to improve the article. All my comments are somehow reflected in the new version. I believe the manuscript has been significantly improved and now warrants publication in JMSE.